# Do Activity Sensors Identify Physiological, Clinical and Behavioural Changes in Laying Hens Exposed to a Vaccine Challenge?

**DOI:** 10.3390/ani15020205

**Published:** 2025-01-14

**Authors:** Hyungwook Kang, Sarah Brocklehurst, Marie Haskell, Susan Jarvis, Victoria Sandilands

**Affiliations:** 1Scotland’s Rural College, Roslin Institute Building, Easter Bush Campus, Midlothian EH25 9RG, UK; marie.haskell@sruc.ac.uk (M.H.); vicky.sandilands@sruc.ac.uk (V.S.); 2Global Academy of Agriculture and Food Systems, The Royal (Dick) School of Veterinary Studies, University of Edinburgh, Easter Bush Campus, Midlothian EH25 9RG, UK; susan.jarvis@ed.ac.uk; 3Biomathematics and Statistics Scotland, James Clerk Maxwell Building, The King’s Buildings, Peter Guthrie Tait Road, Edinburgh EH9 3FD, UK; sarah.brocklehurst@bioss.ac.uk

**Keywords:** vaccine challenge, clinical signs, disease, welfare, sensor technology

## Abstract

Large group sizes and diverse structures in modern housing systems for laying hens can have a detrimental effect on animal health and welfare by limiting health monitoring and the early detection of signs of sickness in individual hens. Wearable sensor technology that can monitor changes in animal behaviour and activity would be beneficial to mitigate these difficulties. This study investigated whether activity sensors detected sickness-type signs caused by side-effects from a live vaccine challenge in laying hens in comparison to the assessment of changes in physical and clinical signs or observing behaviour. Our study showed that wearable sensors detected changes in activity levels, distance travelled and the location of individual hens. Sensor technology to detect changes in behaviour and animal movement may be useful for monitoring animal health and welfare.

## 1. Introduction

Public concern and legislative demands for high-welfare systems for the breeding and housing of livestock animals are continuing to increase around the world [1,2,3], and this has led to the development of housing systems that increase the possibilities for farm animals to express natural behaviours and provide greater opportunities for movement [4,5]. With laying hens, some countries have shifted from housing their flocks in intensive systems to alternative housing systems. However, alternative systems typically house large flocks and often in diverse structures (such as multi-tier units), making it difficult for farmers to assess and monitor the health and behaviour of individual animals [6,7].

The health of animals is an important factor influencing the level of animal welfare, and sick or diseased animals are considered to have poor welfare. Therefore, identifying behavioural changes in sick animals can be a useful indicator in evaluating the welfare of animals [8,9,10]. The early detection of behavioural changes such as decreased activity or decreased feeding in sick animals could improve not only the welfare and health management of animals via the rapid initiation of treatment and lower mortality [11], but also of human health by reducing antibiotic use [12,13], and therefore antibiotic resistance. However, the early identification of sick or diseased animals can be difficult, because prey-species animals, such as chickens, have the tendency to conceal signs of sickness to avoid being spotted by potential predators [7]. In addition, behavioural changes and signs of sickness in the early stages are elusive at the group level [14], and animals are more likely to hide in the housing structure or within flocks [7]. Furthermore, the early detection of changes in animal health could be limited by the subjectivity and inaccuracy of casual observations by farm staff [15,16].

To mitigate these vulnerabilities, efficient and accurate methods that could monitor behavioural changes as an indicator of health and welfare would be beneficial, and the use of automated technology such as wearable sensors that produce data in real time could possibly be utilised to observe individual behaviours and assess the welfare of animals in current housing systems with large groups [3,8,17]. Such sensors have been used to measure feeding and nesting behaviour [18,19], the relationship between feeding behaviour and feeder space [20], space use and activity using an RFID (Radio-Frequency Identification) system [21], jumping and landing forces using a three-axis accelerometer [22] and the locations of individual hens by applying UWB (Ultra-Wideband)-based tags [23]. One consideration with wearable sensors is their reliability; however, several studies have demonstrated that data from sensors agree closely with behavioural observations, with ranges of 85 to >99% accuracy (e.g., [23,24,25]). Therefore, there is evidence that technology can play a role in the accurate remote identification of behaviour patterns, and thus behaviour changes. More studies related to the early detection of disease using sensor technology are required for the health management and welfare of laying hens and the evaluation of future applications of sensor technology in current housing systems. However, in a research context, purposefully infecting animals with disease is ethically questionable, so to test if sensors can identify changes in behaviours related to indicators of disease, using a live vaccine is one alternative. Live vaccines are known to cause side effects similar to disease symptoms in humans [26], pets [27], and cattle [28].

The purpose of this study was to determine if sensors could detect changes stimulated by a live vaccine challenge in laying hens at the same time, or before, they would be identified using traditional clinical sign scoring or observed changes in behaviour. The hypotheses were as follows: (1) When given a vaccine challenge to induce mild respiratory signs, then clinical signs, activity levels and behaviour of hens will change. (2) Sensor technology will identify the changes in activity and behaviour of hens more accurately than the human observation of behaviour will.

## 2. Materials and Method

### 2.1. Animals, Housing and Husbandry

Non-vaccinated pullets (Hy-Line Brown, *n* = 31) were reared from 1 day old to 10 weeks of age at the Roslin Institute (Midlothian, UK). They were individually identified by wing tags. Pullets were then transferred to SRUC’s Allermuir Avian Innovation and Skills Centre (Midlothian, UK) for further rearing to 20 weeks of age. When they arrived at SRUC, all pullets were inspected, and one pullet was culled due to plumage damage. The remaining 30 birds were reared in a home room (approximately 40 m^2^), where they were provided with wood shavings litter on top of a concrete floor, two manually filled, circular feeders, two bell drinkers, two perches (15 cm/bird), pecking enrichment (one pecking block, one alfalfa hay bale, replaced as required), and six nest boxes (from 14 weeks of age). Standard diets (Farmgate rearer pellets and Farmgate layers mash (ForFarmers, Brydekirk, UK)) were provided to the birds according to their age. The birds remained in the home room until they were moved to the experimental room.

The experimental room (Figure 1, approximately 180 m^2^) provided wood shavings litter flooring in the middle of the room (litter zone, 5 m × 7 m), one manually filled circular feeder, one bell drinker, one perch (32.5 cm/hen, 30 cm high), one alfalfa hay bale (replaced as required) and four nest boxes on the litter floor. The rest of the floor, in the outer zone, was bare concrete. Fences were installed in batch 1, day 3 along the walls to prevent hens from entering blind spots in the outer zone where the cameras could not reach. Light intensity in both the home and experimental room was maintained between 15 and 30 lux, and other environmental control and biosecurity followed the management guides for Hy-Line Brown hens and the appropriate SOPs of SRUC.

### 2.2. Treatments and Study Design

A pilot study (unpublished) was conducted prior to the current study to determine which vaccine could induce measurable respiratory and other related signs, and to identify the period of immune response. Briefly, live Infectious Laryngotracheitis (ILT) vaccine (Poulvac^®^, Zoetis, Leatherhead, UK) and Newcastle Disease (ND) vaccine (ND Clone 30, Nobilis^®^, London, UK) were tested using 8 hens (4 hens/vaccine) with two different administration methods (ocular drop, ocular and nasal drop). ILT vaccine using ocular and nasal drop was selected as optimal for producing visible physiological and clinical signs.

For the current study, a total of five batches of four hens were used, each for 12 days in the experimental room. Thus, in total, we used 20 hens over 25–35 weeks of age. The study was conducted in June–August 2022, during which the experimental room mean (daily minimum) temperature was 18.9 ± 1.1 (standard deviation, SD), mean (daily maximum) temperature was 21.3 ± 1.5, and mean humidity was 71.6% ± 6.5. On day 1, four hens (two with mostly white, and two with mostly brown, plumage) were selected from the home room and taken to a bird handling area outside of the experimental room. Both FitBark (FitBark Inc., Kansas City, MO, USA) and TrackLab (Noldus Information Technology, Wageningen, Netherlands) activity sensors were mounted onto the hens’ backs by using a string loop over the hens’ shoulders, which in turn was attached to the sensors using either white or brown insulation tape. Within a batch of four hens, there was therefore one hen per combination: white hen–brown-covered sensors, white hen–white-covered sensors, brown hen–brown-covered sensors, and brown hen–white-covered sensors. Detailed photographs of each side of the hens’ faces were collected. After these procedures, the hens were then introduced to the experimental room (day 1).

On day 3, all four hens were treated with saline, applied by giving two ocular and two nasal drops (one drop in each eye and nostril) as a control. On day 6, ILT vaccine was given to all four hens using the same method. On day 12, the four hens were removed from the room, sensors were removed from each and recharged, and the room was prepared for the next batch. The time between one batch ending and the next batch beginning was 2–5 days.

### 2.3. Data Collection

#### 2.3.1. Physiological Measurements

Physiological measurements were collected by a single assessor (HK). Hens were carefully managed during these measurements to minimise the effects of handling. Respiratory rate (counts/min) and cloacal temperature (°C) of individual hens were recorded twice a day (in the morning and afternoon), on days 1–12. Individual hen weights (kg) (including the weight of the sensors) and feed intake (kg) for each batch, scaled to feed intake per hen per day, were recorded on days 1, 3, 6, 9 and 12 to measure for the change during days 1–3, 3–6, 6–9 and 9–12. Feed intake on day 1 was set at zero.

#### 2.3.2. Clinical Signs

Five clinical signs were scored by a single assessor (HK) on an ordinal scale from 0 to 3, where 0 = normal, 1 = mild, 2 = moderate, 3 = severe (Table 1), twice a day (in the morning and afternoon) from day 1–12 based on the expected main respiratory symptoms of the ILT virus. In addition, the scores of each clinical sign were summed to obtain “total clinical scores”.

#### 2.3.3. Sensor: Activity Level and Distance Travelled

FitBark sensors are 3-axis accelerometers (41 mm × 28 mm × 13.5 mm, weight 10 g) originally designed for pet dogs that measure activity levels (BarkPoints) of individuals in real time. “BarkPoints” are a proxy measure of the animal’s activity, and data were recorded on a minute-by-minute basis and transmitted from the sensor to a mobile application via Bluetooth. FitBark produces an estimate of distance travelled using patterns of activity collected from the accelerometer data. TrackLab is a real-time monitoring system to track behaviours such as distance travelled, location, and speed using UWB sensors (54 mm × 40 mm × 14 mm, 26 g) that are detected by four beacons set in the corners of the room. The expected error in location accuracy is reported to be ±15 cm [29].

FitBark sensors were used to record hens’ activity levels. Distances travelled for each hen were collected by both FitBark sensors (metres/day) and the TrackLab system (metres/hour) in real time. All data were transferred to Excel files for subsequent processing and analysis. The total activity level and/or distance travelled for each hen per day were calculated from each sensor. To avoid biases from incomplete data on day 1 (sensors were attached to hens part-way through the day), data were analysed for days 2–12.

#### 2.3.4. Behaviour and Location

A total of 10 cameras were installed in the experimental room to monitor the behaviour and location of individual hens via human observation (herein referred to as ‘behaviour observations’). Six cameras were installed around the room’s walls, and four were placed in the central litter zone on a post (one facing each wall). Video recordings were continuously collected using the Geovision system (GV-800B) from days 1–12. However, to avoid biases from incomplete data on day 1 (hens entered the room part-way through the day), data were analysed for days 2–12. Each day, scan samples of behaviours and locations were recorded every 15 min for 2 h blocks, three times per day with intervals of at least 2 h between each sampling block during lights on (i.e., between 6:00 and 20:00). The hens’ behaviours and locations were recorded based on an ethogram (Table 2) and behaviours were classified into two categories: active and inactive.

A total of six locations (litter floor zone, outer concrete zone, feeder, drinker, perch, and nest boxes) were recorded to identify the change in location of individual hens based on video scan samples (Table 2). The proportion of time spent by individual hens in each location were also recorded by the TrackLab system, according to pre-set zones (Table 2) (Note that, because TrackLab does not distinguish in the height dimension, the zone around the perch could include on the perch or underneath it).

### 2.4. Ethical Considerations

This study was approved by SRUC’s Animal Experiments Committee (study number POU-AE-03-2022). All hens were rehomed after the experiment.

### 2.5. Statistical Analysis

All data were summarised for each hen by day prior to subsequent statistical analyses. The mean of the twice-daily scores of clinical signs and physiological measurements were calculated for each hen. Counts from video scanning of behaviours and location of individual hens and time they spent in locations from TrackLab were converted to proportions of total daily data for each behaviour/location class per hen. Total daily distance travelled per hen was calculated for TrackLab and FitBark and total mean daily activity per hen for FitBark. There was approximately 4 h of missing data from one bird on day 5 in batch 2, due to the detachment of the pair of sensors; therefore, sensor data from that bird on day 5 were omitted from the analysis. Feed intake data between day 6 and day 9 in batch 2 were omitted because of a large amount of feed scattering by the birds.

Linear mixed models (LMMs) were used to analyse changes over days in the bird weights and feed intake, daily mean physiological measurements, mean clinical sign scores, proportions of each behaviour and each location, activity levels and distance travelled. Clinical scores were analysed from day 7 onwards only, because most scores before day 7 were 0. Residual plots were used to check for normality and homogeneity of variances, and all proportions of behaviour, location and total visit duration of hens in resources and three zones were angular transformed to normalise residuals. Day was included as a categorical variable in the LMMs as a fixed effect, and the random effects were batch, hen and day nested within batch (residual: day nested within hen) (Equation (1)):(1)$$y_{bhd}=\propto +\mu_{d}+\varphi_{b}+\gamma_{h}+\omega_{bd}+\varepsilon_{bhd}\;\mathrm{with}\;\propto \in R,\;\varphi_{b}\sim N(0,\sigma_{\varphi }^{2}),\;\gamma_{h}\sim N(0,\sigma_{\gamma }^{2}),\;\omega_{bd}\sim N(0,\sigma_{\omega }^{2})\mathrm{and}\;\varepsilon_{bhd}\sim N(0,\sigma^{2})$$
where *y* is the response variable, *d* is the day, *b* is the batch and *h* is the hen.

For feed intake, data were measured on a per-batch basis every 3 days, and the random effects were batch (residual: day nested within batch) (Equation (2)):(2)$$y_{bd}=\propto +\mu_{d}+\varphi_{b}+\varepsilon_{bd}\;\mathrm{with}\;\propto \in R,\;\varphi_{b}\sim N(0,\sigma_{\varphi }^{2})\mathrm{and}\;\varepsilon_{bd}\sim N(0,\sigma^{2})$$
where *y* is the response variable, *d* is the day and *b* is the batch.

For ease of interpretation, data are presented rescaled as kg per hen per day, by dividing the estimated means and SEs from the LMM by 12 (4 hens/batch × 3 days). *p* values for the effect of day were based on approximate *F* tests using the Satterthwaite method for denominator degrees of freedom. Graphs show daily data mean ± SD, and a table of means ± standard errors (SEs) and standard errors of differences (SEDs) for the day effect estimated by LMMs is given in the Appendix A. Tukey’s HSD test for pairwise comparisons of estimated means was used in some cases to corroborate reported trends over days. *p* values were considered to be significant when <0.05. Pairwise relationships between daily measurements were explored using Pearson’s correlation coefficient and scatterplot matrices. LMMs were used to examine pairwise associations between sensor measurements and mean total clinical scores (TCSs) (in this case, from days 2–12) with the random effects batch, hen, day and day nested within batch (residual: day nested within hen) (Equation (3)):(3)$$y_{bhd}=\propto +{\beta x}_{bhd}+\varphi_{b}+\gamma_{h}+\omega_{bd}+\varepsilon_{bhd}\;\mathrm{with}\;\propto ,\beta \in R,\;\varphi_{b}\sim N(0,\sigma_{\varphi }^{2}),\;\gamma_{h}\sim N(0,\sigma_{\gamma }^{2}),\;\omega_{bd}\sim N(0,\sigma_{\omega }^{2})\mathrm{and}\;\varepsilon_{bhd}\sim N(0,\sigma^{2})$$
where *y* is the response variable, *x* is the TCS, *d* is the day, *b* is the batch and *h* is the hen.

Variance components were compared between LMMs with no fixed effects and with the fixed effect TCS as a numerical variable to elucidate whether, for example, the extent to which sensor measurements were explaining differences in TCS between times overall and within birds, or between birds overall and within times and so on.

Data processing and statistical analysis were performed in Excel and in the R system for statistical computing (version 4.4.1 ([30]) accessed from RStudio 2024.04.2 Build 764 [31]). LMMs and the analysis of results were carried out using R packages lmer4 [31], lmerTest [32], and emmeans [33]. A data dictionary explaining all analysed variables is given in Appendix A.

## 3. Results

All 20 birds completed their 12 days in the experiment. A total of 480 assessments were conducted for clinical signs and physiological factors. In the 5280 observations of hens’ behaviour via video scanning there were 76 missing data points (1.4%), mainly due to blind spots identified in batch 1. Sensor data were collected automatically and in real time throughout the entire experiment.

### 3.1. The Effect of Day on Physiological Measurements and Clinical Signs

There were significant effects of day on feed intake, body weight, cloacal temperature and the scores of some clinical signs (Table 3 and Appendix A).

Feed intake values from day 6–9 for batch 2 birds were ignored, due to feed spillage. Mean feed intake (kg) over days at the batch level was marginally significantly different between days (*p* = 0.031), as it increased from day 1–3 to day 3–6, plateaued between day 3–6 and day 6–9 and then decreased between day 6–9 and day 9–12 (Figure 2a); the figure shows feed intake rescaled to per hen per day, to aid interpretation. Mean body weights changed significantly over days (*p* < 0.001) as they decreased between day 1 and day 3 and again between day 9 and day 12 (Figure 2b). Mean body weights were significantly lower on day 12 than on day 1. Mean cloacal temperatures were significantly different between days (*p* < 0.001), with an increase on day 11 compared to days 4–5 and days 7–9 (Figure 2c). There were no significant differences in mean respiratory rates between days (*p* = 0.179).

Clinical scores before day 6 were all 0, and most were 0 on day 6. Mean scores of depression (*p* = 0.009) and ocular discharge, conjunctivitis and total clinical scores (all *p* < 0.001) from day 7 onwards were significantly affected by day (Table 3 and Appendix A). The biggest effects (as evidenced by the size of the F statistics) for individual clinical signs were in ocular discharge (OD) and conjunctivitis (CON), which both increased from day 6 (i.e., the day of vaccination) onwards (Figure 3a). Hens showed higher OD scores on days 10–12 than on any other day and had higher CON scores on days 8–12 than on days 1–6. Mean scores of depression (DEP) were highest on the last study day (Figure 3b). Total clinical scores showed a consistent rise after vaccination on day 6 (Figure 3a).

### 3.2. The Effect of Day on Sensor Measurements—Activity and Distance

There were significant effects of day on all activity and distance sensor measurements (Table 4 and Appendix A).

The mean activity levels of hens as recorded by the FitBark sensors were significantly different between days (*p* < 0.001) (Figure 4a). Mean activity levels gradually declined over days 3–10, with a sharper decline on day 11, but with some recovery on day 12. Hens showed the lowest mean activity level on day 11. The mean distances travelled by hens were significantly affected by day (both FitBark and TrackLab sensors, *p* < 0.001). The mean distances travelled according to FitBark showed an almost identical pattern to the activity levels according to FitBark, with distances travelled declining gradually from day 3 to day 10, with a sharper decrease on day 11, and a rise at day 12 (Figure 4b). The mean distances travelled according to TrackLab were substantially less than that recorded by FitBark, and with a less pronounced decline in distance over days, which was fairly steady from days 3 to 12. Hens covered the least distance on day 11 with FitBark, and day 12 with TrackLab (Figure 4b).

The variance components for hens were substantial in both sensors but much larger for TrackLab than for FitBark. The batch and batch-by-day variance components were also large for TrackLab but near zero for FitBark.

### 3.3. The Effect of Day on Behaviours from Video Scans

There were significant effects of day on some behaviours, although most of these were fairly marginal (Table 5 and Appendix A). Feather pecking and aggressive interactions were not observed.

The proportions of sitting (*p* = 0.044), foraging (*p* = 0.036), and feeding (*p* = 0.004) behaviours were significantly affected by day. The changes in foraging behaviours did not show obvious tendencies before and after vaccine challenge at day 6, and sitting behaviours increased consistently after day 9 (0.012 at day 9 to 0.062 at day 12, Figure 5). Hens steadily decreased the proportions of feeding from day 8 to day 12 by over 50% (0.12 at day 8 to 0.04 at day 12). Active behaviours were largely consistent between day 3 and day 8, with a small increase at day 9 and then a decrease at day 10. There were no statistically significant differences between days in other behaviours.

### 3.4. The Effect of Day on Hen Location from TrackLab and Video Scans

The proportions of total visit duration to the litter zone (*p* < 0.001) and perch (*p* = 0.037) from TrackLab were significantly different between days (Table 6 and Appendix A).

Hens decreased the mean proportions of time spent in the litter zone from days 2 to 7 after which the mean proportion was fairly stable (Figure 6a). The mean proportion of time spent at the perch increased from days 2 to 6 and remained fairly stable from days 7 to 12 (Figure 6b). There were no statistically significant differences between days on the proportions of time spent at the inner (and outer) zones, drinker, feeder, and nest boxes.

The proportions of visit counts of hens in the litter zone from video scanning were also significantly different between days (*p* = 0.012); however, in contrast to TrackLab, there were no observations of hens at the perch from video scanning (Table 7 and Appendix A).

There was a decline in the mean proportions of visit counts of hens in the litter zone from days 2 to 5, after which the mean proportion was fairly stable (Figure 7). The mean proportion of visit counts of hens in the litter zone from video scanning is higher than the mean proportion of the total visit duration to the litter zone recorded by TrackLab (Figure 6a), but that is as expected since TrackLab was recording for the full 24 h, including lights off when hens are likely to be on the perch, whilst video observations were intermittent and during daylight hours only.

### 3.5. Correlations

The correlation matrix between all variables is shown in Appendix A. With regard to the physiological measurements and clinical signs, the analysis showed that there was a weak correlation between feed intake and weight (*r* = 0.23). For the clinical sign scores, there was a strong positive correlation between conjunctivitis and ocular discharge (*r* = 0.88). Depression was also correlated with conjunctivitis (*r* = 0.54) and ocular discharge (*r* = 0.54), and abnormal breathing was correlated with conjunctivitis (*r* = 0.55). Total clinical score was most strongly influenced by conjunctivitis (*r* = 0.96) and ocular discharge (*r* = 0.92).

For the behaviours, a negative correlation was found between feeding and foraging proportions (*r* = −0.46). It is interesting to note (from Figure 8) that this trade-off between feeding and foraging is a reflection of how hens spend their time daily; the scatter plot shows how the negative correlation occurred on individual days (not as a result of between-day differences) and within-day correlations show that this relationship breaks down around days 6–7 and 10–11. The proportion of active behaviours was negatively correlated with total clinical score (*r* = −0.38), as was feeding and total clinical score (*r* = −0.22), particularly on day 11 (*r* = −0.69, −0.63, respectively) when it was also negatively correlated with temperature (*r* = −0.44, −0.45, respectively).

For the hen location, the proportion of total visit duration to the feeder (from TrackLab) was positively correlated with the feeding location and behaviour proportions from video observations (*r* = 0.62, 0.57, respectively), and feed intake (*r* = 0.30). There was a positive correlation between the proportion of visit counts of hens in the litter zone (from video observations) and the proportion of total visit duration to the litter zone (from TrackLab) (*r* = 0.64), a relationship which is evident both between and within days (Figure 8), though it varies between days and breaks down on day 10. The proportion of total visit duration to the litter zone (from TrackLab) was negatively correlated with total clinical score over all days (*r* = −0.37), but especially on day 11 (*r* = −0.55).

For the sensor data, there was a very strong positive correlation between activity level and distance travelled from FitBark (*r* = 0.98), and a positive correlation between distance travelled from FitBark and TrackLab (*r* = 0.55), which is because of both within-day and between-day changes (Figure 8). Graphs of longitudinal data for each hen over days (Appendix A) show how, apart from the change in scale, the activity and distance travelled from FitBark are almost identical. The positive association between activity (and hence distance travelled) from FitBark (r = 0.54) with distance travelled from TrackLab appears (Figure 8) to break down on days 9–11 when activity was low (see individual bird data in Appendix A). These longitudinal plots show that there was a general decrease over time for all the sensor measures but there was much variation between birds as well as local variation within birds, with some clear outliers. It was apparent that one FitBark sensor (tag 1), which was worn by one bird per batch (bird_id: 1971, 1981, 1975, 1936, 1992) consistently recorded marginally greater distances travelled relative to given daily activity than other FitBark sensors (see Appendix A). Clearly, distance travelled is estimated as a perfect linear relationship to daily activity, which for some reason varied slightly for tag 1.

The proportions of active behaviours were positively correlated with activity level and hence distance travelled from FitBark (*r* = 0.46, 0.44, respectively), and with distance travelled from TrackLab (*r* = 0.37), relationships which are evident both between and within some, but not all, days (Figure 8). Distance travelled from TrackLab was positively correlated with the proportion of walking (*r* = 0.57) and negatively correlated with preening (*r* = −0.42), but these correlations were not so apparent for FitBark. Thus, whilst some of the behaviours show fairly similar correlations with measurements from the two sensors (e.g., sit, run, feed, forage) others (e.g., stand, walk, preen) differ between FitBark and TrackLab.

The total scores of clinical signs were negatively correlated with activity level from FitBark (*r* = −0.40) and the scatter plot (Figure 8) suggests this was largely as a result of between-day changes, though negative correlations are evident later when clinical scores are higher, indicating that birds with higher clinical scores were less active according to video observations (r = −0.69 on day 11) and had low activity levels according to FitBark (r = −0.49 on day 11). The correlations of clinical score measurements with activity measured by FitBark were slightly stronger than the correlations with distance travelled measured by Tracklab. The longitudinal plots (Appendix A) show how clinical scores clearly increase in most birds in response to the vaccine, but also how the trend over time differs from that seen in the sensor measurements. The peak correlation occurring at day 11 is when activity level from FitBark was generally low for all birds.

### 3.6. The Effect of Total Clinical Scores on Sensor Measurements—Activity and Distance

From the LMMs (Table 8), both the activity level measured by FitBark and the distance travelled measured by Tracklab are highly significantly related (*p* < 0.001) to the total clinical scores (TCSs), both having negative estimated gradients indicating that both sensor measures decreased with increasing total clinical scores. The relationship is slightly stronger for FitBark than for Tracklab (as evidenced by the larger F statistic for FitBark). Comparing the variance components in the LMMs with and without TCS in the fixed effects shows that TCS mostly explains differences between days, though the variances between batches and between days within hens (the residual) also reduce, particularly for FitBark. Interestingly, the hen variance components increase for both measurements, suggesting that, having adjusted for total clinical scores, inherent differences in hens’ activity and distance travelled persist.

## 4. Discussion

This study was designed to investigate physiological and behavioural changes in adult laying hens in response to a vaccine challenge through clinical scoring, behaviour observations and sensor technology. We found that, while clinical signs increased after a vaccine challenge, changes in other measures were less clearly related to the challenge and may have been due to effects over time alone, or in combination with the vaccine.

### 4.1. The Effect of Day on Physiological Measurements and Clinical Signs

In the current study, mean cloacal temperatures were significantly affected by day; however, they were within normal ranges (40.7–42 °C): normal body temperature ranges for poultry are 41–42 °C, while a temperature above 42 °C is considered a fever [34,35]. While fever response and behavioural changes such as anorexia in animals are considered a means of survival to overcome acute infectious diseases [36], this may suggest that monitoring cloacal temperatures is not suitable as an early indicator of sickness signs or of some acute respiratory diseases in laying hens. The mean body weight of hens had significantly declined by day 9, 3 days after vaccination, whereas feed intake had decreased (after a presumed period of adjustment between days 1 and 3) by day 10–12. This result is similarly consistent with other studies on the physiological effects of disease [37]. We expected a strong correlation between feed intake and body weight; however, only a weak correlation (*r* = 0.23) between them was found, with weight loss occurring despite increased feed intake before vaccination. The weight loss that occurred, despite increased feed intake, might be expected if there were a commensurate increase in active behaviours and distance travelled during the study; however, the performance of active behaviours (as observed from video scanning) was generally consistent between days 3 and 8, with a decrease later on in the study. Instead, it may be that feed intake was a combination of feed consumed and feed spilled by birds, as they could scrape feed out of the feeder and into the litter.

It has been demonstrated that clinical signs in poultry with respiratory diseases were mainly observed in the eyes, which is associated with the immune system’s lymphoid tissues in the eyes and upper respiratory tract [38,39]. In this study, mild to severe clinical signs in the eyes of all 20 hens were identified. A strong positive correlation between ocular discharge and conjunctivitis (*r* = 0.88) was identified and, in most cases, hens showed conjunctivitis followed by ocular discharge. Research by Oldoni and colleagues [40] demonstrated that clinical signs such as mild conjunctivitis and depression were observed in hens inoculated with live ILT vaccines using ocular and nasal drops. Similarly to the results of their study, here, hens with severe conjunctivitis also showed swelling of the conjunctiva and shut eyes as well as accompanying moderate depression, with birds adopting a crouching posture. A total of eight hens showed abnormal breathing, mainly mild indicators like head shaking, and moderate abnormal breathing such as intermediate open mouth breathing was recorded in eight hens. There were no occurrences of severe (score 3) abnormal breathing or depression, which appeared to be related to the characteristics of a live attenuated vaccine inducing mild immune responses. Total clinical scores increased consistently after the vaccine challenge, in particular showing a sharp increase between days 9 and 10. This showed a similar duration of the immune response to the ILT vaccine as described in the vaccine’s data sheet (i.e., 4 to 5 days post vaccination) [41].

### 4.2. The Effect of Day on Activity Sensor Measurements

The current study suggested that the two sensor technologies detected decreases in activity/distance travelled that could have been caused by the vaccine challenge in the six days after the vaccine was applied. The effects on activity and moving distance in animals caused by disease or signs of sickness have been investigated in other studies, and the results were similarly consistent with those of this study. However, activity was declining during the entire study and not just after day 6 (when the vaccine challenge was given). In laying hen studies, hens infected by pathogens such as Salmonella showed suppressed activity [2,42] and decreased activity in SPF (specific pathogen free) chickens infected with the High Pathogenic Avian Influenza (HPAI) virus was detected by accelerometers [43]. With broilers, sensors were applied to detect changes in moving distance and it was demonstrated that broilers with lameness and high gait scores showed a decrease in activity level according to their age [44] and over time [45]. However, one drawback to using sensors in commercial flocks is that only a sample of birds can be monitored, due to their cost. Due to variation in individual bird behaviour, it would be important to monitor ‘normal’ individual behaviour and then look for changes per bird, to give an indication of a disease emerging, for example. In addition, it would be difficult to routinely catch a few birds instrumented with sensors in a large flock, which would be necessary for battery charging, sensor checks, monitoring birds for signs of irritation from sensor attachment and so on.

Here, one unexpected finding was that there were significant decreases in activity level and distance travelled after saline was given on day 3, before the vaccine challenge had taken place. Given previous results demonstrating that the attachment of sensors has no significant effect on clinical scores for foot pad, hock dermatitis and gait score in broilers [46] or on behaviours such as preening in laying hens [17] compared to hens without sensors, we felt it was unlikely to be an effect of the sensors themselves. It is possible that the transfer of the hens into the spacious experimental room from the smaller home room led to a higher-than-normal activity and distance travelled for hens in the early period (from days 1 to 3) which then gradually returned to normal ranges after an adequate habituation period. In future studies, it is suggested hens should be allowed to acclimatise for longer until the impact of the novel environment and/or increased room space on activity and movement plateaus to a consistent level. Another possibility is that the saline itself resulted in decreased activity/distance travelled. Such effects could have been ruled out if we had controlled for the order of the saline and vaccine, however because the duration of any vaccine effects was unknown, we chose not to do this.

We did not validate either sensor because these have been verified previously in other studies (FitBark: [47,48]; TrackLab: [45,49]). However, we discovered that the data from one FitBark sensor showed a slightly different relation between activity and distance travelled than the other three. This implies the settings of each sensor were not identical even though they are manufactured in the same way, or that one of our tags was faulty in some way, which introduced minor errors. Unless the FitBark can be calibrated by the user, this is problematic. There were large differences in mean distances travelled (metres) between the two sensors, but they were positively correlated (*r* = 0.55) and showed similar decreasing patterns over days. The strong correlation between mean distance travelled and activity level for FitBark (*r* = 0.98) is presumably due to daily distance travelled being estimated directly, using a simple scaler multiplier, from daily activity levels measured by accelerometers alone, as FitBark does not contain sensors of actual location. For TrackLab, the data are measured by signal transmission between each sensor and four beacons within the experimental room in real time, so it would be expected that the TrackLab system would be more accurate than the FitBark sensor at measuring distances. Thus, the correlation (*r* = 0.55) between distances travelled from the two sensors likely just reflects that true distance travelled (measured more accurately by TrackLab) is correlated with true activity (measured by FitBark) (*r* = 0.54).

The variance components for daily distance travelled from TrackLab were very large for batches and hens, and large for days within batches, whilst the variance component for the daily activity of hens from FitBark was also fairly large but substantially smaller, by about one-third, than for Tracklab. Data collected from one FitBark sensor differed from those of the other three FitBark sensors. All sensors were used repeatedly during the five replicated batches, which could have contributed to the large variance components for hens, particularly for TrackLab, (as sensor was not included in the random effects in the LMMs). It is not possible to judge whether the larger variance components for Tracklab are as a result of data inaccuracies or as a result of genuine variability in distance travelled by hens. Similarly, it is possible that FitBark smoothed out genuine noise in the activity data, resulting in increased evidence for the day effect. Either way, these large hen variances, which persisted in models examining sensor measurements in relation to total clinical scores, have important implications for the way in which sensors should be used to detect health and welfare problems in individual birds in real time. A large variance component for individual hens suggests that detecting problems in individual hens would be optimised by observing changes in each hen relative to their past normal activity. Large variance components for batch or batch by day (seen here for Tracklab only), if accurate, are indicative of group-level differences overall and differences between experimental days within groups. This suggests that to use these sensors in real commercial flocks, flock-wide changes (for example, due to management), should also be adjusted for. Whilst this could be achieved in an automated way based on sensor data, such as adjusting individual sensor data for current flock-wide means, such changes, when they occur, would also need to be flagged in real time to check that they do indeed coincide with flock management. This is so that changes indicative of flock-wide problems (such as disease outbreak or the failure of environmental control or automated feeders or drinkers) are not missed.

The activity measured by FitBark appears to be slightly more sensitive to the vaccine challenge compared to distance travelled measured by the more expensive TrackLab. This is suggested by the larger F statistics for the day effect and for the effect of total clinical scores as well as larger reductions in variance components, and lower negative correlation with clinical scores, for daily activity measured by FitBark. It is unclear, however, whether this is because true activity is better associated with the vaccine challenge than true distance travelled or whether this is impacted by the inaccuracy of the measurements of these quantities by the sensors. In addition, it is noteworthy that the overall picture from both sensors is that the average activity and distance travelled generally declined over the entire study, even prior to the vaccine challenge at day 6.

### 4.3. The Effect of Day on Behaviours

Animals in the early stages of sickness or disease show changes in their behaviour such as decreased feeding, drinking and activity [14]. The results of the current study, in which the proportion of active behaviours was somewhat reduced three days after the vaccine challenge, support this statement, albeit only mild changes were seen, most likely because this was a vaccine response and not ill health. In addition, (and true of all behaviours observed here from videos), more accurate records of active behaviour (and thus any real changes) may have been gathered if data recording from videos had covered a longer period of lights on; here, we observed three blocks of 2 h of lights on (out of a total of 14 h of lights on).

In wild conditions, Red Junglefowl allocate much of their time to foraging activities [50], and Dawkins demonstrated that they spend a large proportion of time foraging (ground pecking and ground scratching) in semi-wild conditions [2,51]. In the current study, foraging was the most prevalent behaviour, accounting for 41% of all observed behaviours via video scanning. Significant changes over days were identified in foraging behaviour, but no clear pattern was observed. This may suggest that foraging behaviour is a highly instinctive and essential behaviour for survival. The proportion of feeding behaviour steadily decreased 2 days after the vaccine challenge (from days 8 to 12), which may account for the weight loss seen in the latter half of the study. The increase in sitting behaviour between days 9 and 12 appears to be related to the aggravation of clinical signs. This is partially in accordance with other studies reporting a decline in feeding and an increase in sitting behaviour according to disease infection. Laying hens infected with Salmonella showed decreased feeding on pellets and waxworms [2]. Laying hens injected with lipopolysaccharide (LPS, bacteria’s cell membrane) to induce signs of sickness revealed increased static behaviours such as sitting and sleep-like posture and decreased walking in a free-range housing system [42].

Contrary to expectations and the results of other studies [14,36], there were no significant changes in the proportion of drinking behaviour. In the current study, data on water consumption were not collected but it is suggested that future studies should investigate this. Pecking behaviour toward sensors of different colours to their feather colour (used to distinguish individuals) was rarely observed during video scanning and assessments for physiological measures and clinical scoring. Pecking toward hens and flapping behaviours were rarely observed, and perching was never observed, throughout the whole experimental period, and this indicated the limitation of this method of behaviour observation for tracking subtle changes in infrequent or short-duration behaviours.

### 4.4. The Effect of Day on Hen Location from TrackLab and Video Scan

In the current study, the proportion of the total visit durations of individual hens to four resources and three zones using the TrackLab system were investigated over days. In other studies using UWB tags to track the activity and location of chickens and to investigate the accuracy of sensors through comparison with behaviour observations via a laser distance meter [46] and a video analysis system [23], both sensors resulted in a high level of accuracy of 85%.

We expected (1) hens to spend most of their time in the inner zone prior to vaccination and (2) hens with clinical signs to spend more of their time in the outer zone. Hens were observed via TrackLab to spend their time in the inner zone during the day, with the proportion of total visit duration ranging from 0.37 to 0.99. However, there were no significant differences in total visit duration for the inner and outer zones before and after the vaccine challenge evident via the TrackLab system (*p* = 0.84) or from video scanning (*p* = 0.06). This appears to be due to a high degree of variation in data depending on the differences in the individual responses of each hen. Significant differences in total visit durations for the litter zone and perch were identified over days with TrackLab, but the large changes were only observed before the vaccine challenge (a sharp decrease for the litter zone, a moderate increase for the perch). These unexpected findings before the vaccine challenge may suggest that hens needed more time to habituate to the new environments before returning to their routine patterns of normal behaviour. After the vaccine challenge, the proportion of total visit duration to the litter zone slightly decreased on day 7, plateaued between days 8 and 10 and then decreased sharply on day 11. There was no perch use observed via TrackLab during the entire period of lights on (which corroborates what was seen in behaviour observations from videos), and the proportion of total visit duration for the perch remained stable between days 7 and 11.

Total visit durations to resources from the TrackLab system do not necessarily mean that hens used them. For example, results through video scanning showed that the total number of visits to the feeder location (i.e., within 30 cm of the food hopper) was 727 and actual number of uses of the feeder (feeding behaviour) was 526 (72% of location). Therefore, data of total visit durations to resources collected from TrackLab have some limitations in tracking changes in behaviours such as feeding and drinking. When comparing the hen location data from video scanning and TrackLab, there was a positive correlation between the proportion of visit counts of hens via video scanning and total visit duration from TrackLab for the feeder (*r* = 0.62), the inner zone (*r* = 0.51) and the litter zone (*r* = 0.64). However, weak correlations at the drinker (*r* = 0.29) and nestbox (*r* = 0.13) were identified. These differences may reflect the limitations of scan sampling compared to continuous observation through the entire video recording. Additionally, observations of hen location using video scanning (both during the set 2 h × 3 period recording phases, but also anecdotally, on reviewing all video footage as a data check against TrackLab) showed that no hens used the perch at all during the entire period of lights on, but the TrackLab system classified hens that were within 30 cm of the perch (including underneath it) as in the perch zone. Therefore, this proximity to the perch resulted in data classed as visit durations to the perch. This demonstrates the limitation in accuracy of TrackLab, which cannot distinguish location in the vertical plane.

## 5. Conclusions

To our knowledge, this is the first study to examine changes in behaviour, physiological measures and clinical signs induced by a vaccine challenge using sensor technology on individual laying hens. We found that a live vaccine challenge was successful in inducing immune-response side effects, causing physiological and behavioural changes in individual hens. It was beneficial from an animal welfare perspective to induce changes in behaviour that could be recorded by sensors without instigating actual disease. However, some changes in activity, particularly prior to the vaccine challenge, appeared to occur with time alone, which might be better elucidated with a longer acclimatisation period in the experimental room before the vaccine challenge. Despite this, measurements from both sensors were strongly negatively associated with total clinical scores, indicating that sensors identify changes in activity and/or distance travelled that can potentially be related to side effects which are similar to signs of disease in animals. Sensors might be useful as a potential enhancement to human observation to detect changes in activity and movement in individual hens as an indicator of health or welfare problems, provided that individual hen behaviours are characterised prior to any disease outbreak, and consideration of the management of hens with sensors and the technology themselves are factored in. Sensors may have further use in commercial poultry systems, by monitoring resource use, alerting producers to subtle changes in their behaviour (for example, predicting smothering events), and predicting the effects of stressful events such as aggressive feather pecking and keel bone damage. Future studies could test the use of sensors in animals likely to develop disease, and in larger groups, as further validation of their ability to detect changes in those circumstances. For sensors to be practical in a commercial environment, they must be smaller, less expensive, easily applied and robust, with a long-life or self-charging power source that does not need recharging within a flock’s lifetime. Sensors that are similar (in size and application) to wing tags or leg rings would be pragmatic. In addition, data must be easy to access and interpret for farm staff. Sensor technology will continue to contribute to Smart Farming and Precision Agriculture, ideally with animal welfare at its core.

## Figures and Tables

**Figure 1 animals-15-00205-f001:**
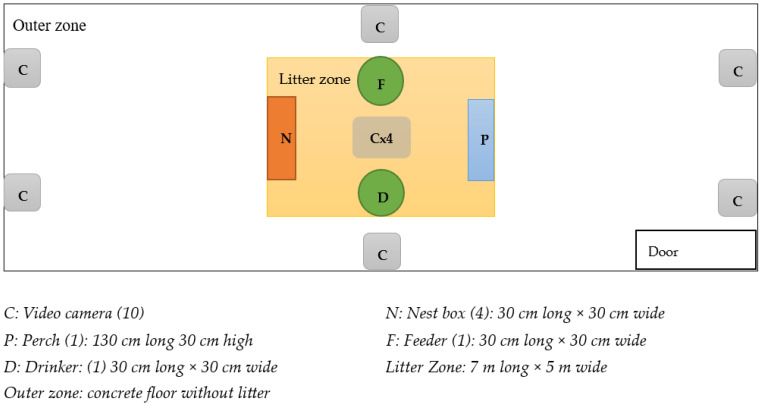
The floor plan of the experimental room and the furniture included there. The number of each item is given in brackets. (Figure is not to scale).

**Figure 2 animals-15-00205-f002:**
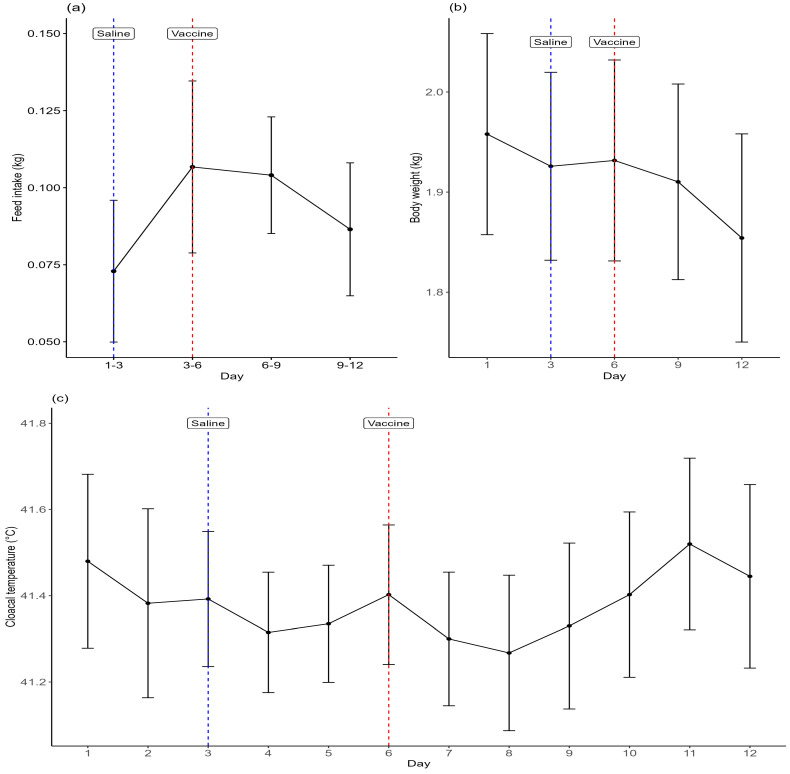
Mean (±SD) values of (**a**) feed intake (kg) rescaled to show per hen per day, (**b**) mean individual body weight, and (**c**) mean cloacal temperature. Days on which saline and live vaccine were administered to birds are indicated.

**Figure 3 animals-15-00205-f003:**
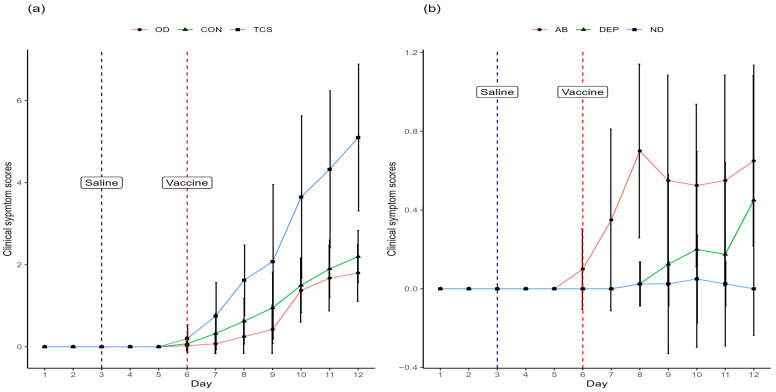
Mean (±SD) clinical sign scores of (**a**) ocular discharge (OD), conjunctivitis (CON), mean sum scores of all five clinical signs (TCS) and (**b**) abnormal breathing (AB), depression (DEP), nasal discharge (ND). Days on which saline and live vaccine were administered to birds are indicated.

**Figure 4 animals-15-00205-f004:**
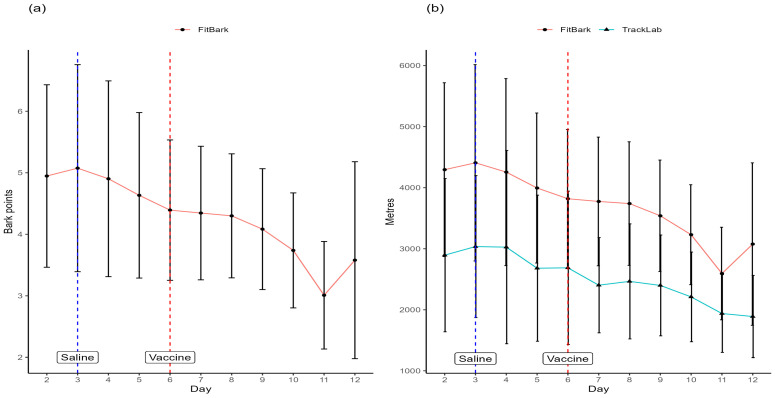
(**a**) Mean (±SD) activity level (in ‘Bark points’) recorded by FitBark and (**b**) mean (±SD) distance travelled (in metres) recorded by FitBark and TrackLab. Days on which saline and live vaccine were administered to birds are indicated.

**Figure 5 animals-15-00205-f005:**
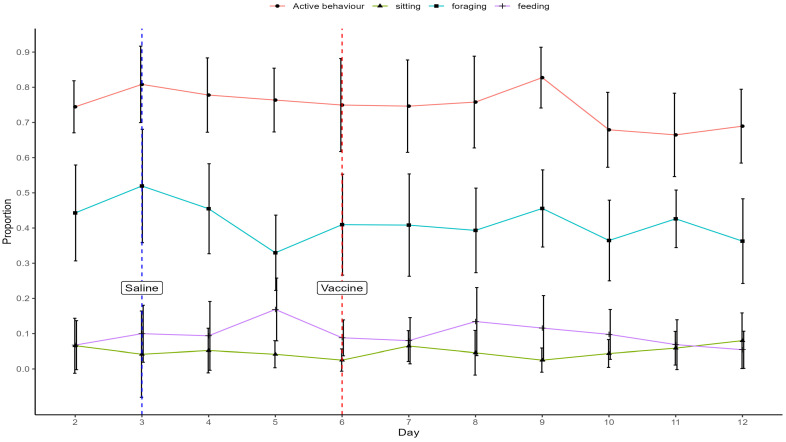
Mean (±SD) proportions of active behaviours, sitting, foraging, and feeding. The days on which saline and a live vaccine were administered to birds are indicated.

**Figure 6 animals-15-00205-f006:**
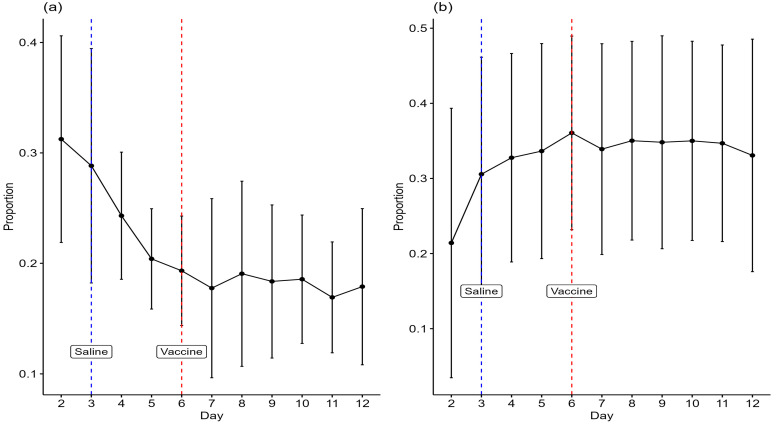
Mean (±SD) proportions of time spent in the (**a**) litter zone and (**b**) at the perch, according to TrackLab. The days on which saline and a live vaccine were administered to birds are indicated.

**Figure 7 animals-15-00205-f007:**
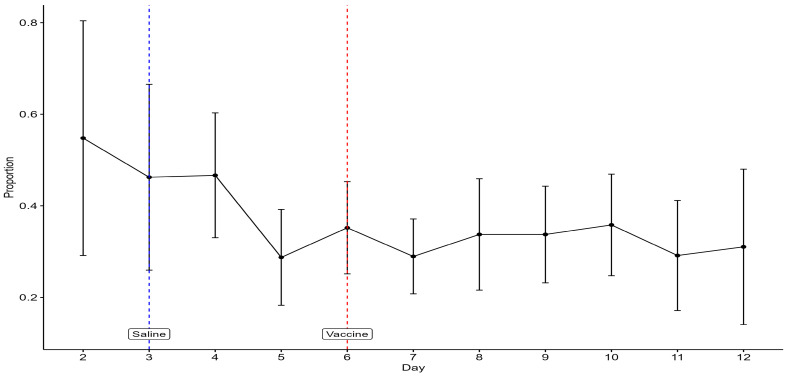
The mean (±SD) proportions of visit counts of birds in the litter zone by video scanning. The days on which saline and a live vaccine were administered to birds are indicated.

**Figure 8 animals-15-00205-f008:**
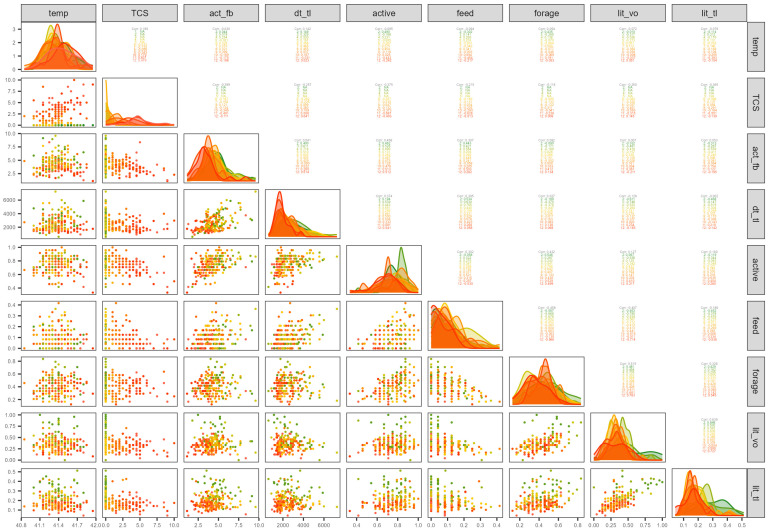
A scatterplot matrix between variables, coloured by days 1–12 (colour scales for day: 
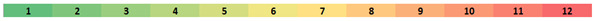
). The curves on the diagonal show the distribution for each variable on each day. Pearson’s correlation coefficients are shown in the top right between pairs of variables, over all days and on each day. The variables shown in the figure are the mean cloacal temperature (temp), total score of clinical signs (TCS), mean activity level from FitBark (act_fb), mean distance travelled from TrackLab (dt_tl), proportions of active behaviour (active), feeding (feed), foraging (forage) behaviour, proportions of counts in the litter zone (lit_vo) from video scans and proportions of total visit duration to the litter zone (lit_tl) from Tracklab.

**Table 1 animals-15-00205-t001:** Scoring system of clinical signs.

Score	Abnormal Breathing	Nasal Discharge	Ocular Discharge	Conjunctivitis	Depression
0 (Normal)	Normal breathing, no breathing noise	Normal both nostrils without nasal discharge	Normal both eyes without eye discharge	Normal conjunctiva	Normal activity and posture
1 (Mild)	Unusual breathing noise which may include sniffling, sneezing, snorting, head shaking	Nasal discharge from one nostril (accompanied by clear and serous exudate)	Ocular discharge from one or both eye(s) (accompanied by clear and serous exudate)	Redness of conjunctiva (one or both eyes)	Sitting with ruffled feather
2 (Moderate)	Intermittent open mouth breathing, mild dyspnoea, can also include signs seen in (1)	Nasal discharge from both nostrils (accompanied by clear and serous exudate)	Ocular discharge from both eyes (accompanied by mucous and purulent exudate)	Moderate conjunctivitis with swelling of conjunctiva, swollen eyelid(s)	Sitting hunched up and crouching posture with ruffled feathers, eyes partially closed
3 (Severe)	Continuous open mouth breathing, severe dyspnoea with cyanosis of comb and skin, can also include signs seen in (1)	Nasal discharge from both nostrils (accompanied by mucous and purulent exudate)	Ocular discharge from both eyes (accompanied by mucous and purulent exudate)	Severe conjunctivitis with swelling of conjunctiva and eyes shut	Lying hunched over and crouching posture with droopy wing, head tucked in towards the body, eyes fully closed

**Table 2 animals-15-00205-t002:** Ethogram of behaviours and locations recorded from video, and locations from Tracklab.

	Description	Definition (Behaviours from Video)
Inactive	Standing	The hen is standing on the floor or the structure.
	Sitting	The hen is sitting on the floor or the perch with eyes open or closed.
	Nesting	The hen is inside the nest box.
Active	Dustbathing	The hen sits in the litter, while scraping a bowl with her foot and rubbing her wings and head into the litter; she may be showing wing tossing to throw the litter onto her back.
	Walking	The hen is moving over the ground on her feet.
	Running	The hen is moving rapidly over the ground on her feet (i.e., faster than walking).
	Foraging	The hen is pecking at and/or scratching at the litter.
	Feeding	The hen has her head in the feeder and is pecking at feed with her beak.
	Drinking	The hen is dipping her head into the water trough, tipping her beak back to swallow.
	Flapping	The hen is moving her wings up and down.
	Preening	The hen is cleaning and arranging her feathers using her beak.
	Feather pecking	The hen is pecking at and/or pulls at the feather(s) of another hen, often repetitively.
	Pecking (resources)	The hen is pecking at feeders (away from the feed), drinkers (away from the water), or resources (walls, perches, nest boxes).
	Aggressive interaction	The hen shows agnostic pecking, clawing, leaping, fighting towards another hen (usually towards the head or neck).
	**Description**	**Definition (Locations from Video and TrackLab)**
Video	Feeder	The hen is within 1 bird length (30 cm) of the feeder.
	Drinker	The hen is within 1 bird length (30 cm) of the drinker.
	Perch	The hen is on the perch rail.
	Nest box	The hen is inside of a nest box.
	Litter zone	The hen is on litter only, but not near the feeder, drinker, perch or nest box.
	Inner zone	The sum of litter zone + drinker + feeder + nest boxes + perch.
	Outer zone	The concrete zone (i.e., excluding the inner zone).
TrackLab	Feeder	The feeder and the space within 1 bird length (30 cm) from the feeder.
	Drinker	The drinker and the space 1 bird length (30 cm) from the drinker.
	Perch	On or under the perch rail and the space 1 bird length (30 cm) from the perch rail.
	Nest box	The nest box and the space 1 bird length (30 cm) from the nest box.
	Litter zone	The hen is on litter only, when not near the feeder, drinker, perch or nest box.
	Inner zone	The sum of litter zone + drinker + feeder + nest boxes + perch.
	Outer zone	The concrete zone except for the inner zone.

**Table 3 animals-15-00205-t003:** Tests from LMMs of effect of day on physiological measurements (i.e., feed intake, body weight, cloacal temperature and respiratory rate) and clinical signs (i.e., abnormal breathing, ocular discharge, nasal discharge, conjunctivitis, depression). All scores for each clinical sign were also summed to give total clinical scores. Columns show numerator degrees of freedom (ndf), denominator degrees of freedom (ddf), F statistics and *p* values. All variables were analysed daily, apart from feed intake, which was measured per batch at four intervals, 1–3, 3–6, 6–9 and 9–12 days; body weight, which was measured per bird on days 1, 3, 6, 9 and 12; and clinical scores which were analysed daily from days 7 to 12 only.

	Effect of Day			
	ndf	ddf	F Statistic	*p* Value
Feed intake	3	11	4.28	0.031
Body weight	4	16	14.39	<0.001
Cloacal temperature	11	209	4.94	<0.001
Respiratory rate	11	44	1.46	0.179
Abnormal breathing (AB)	5	20	0.68	0.647
Ocular discharge (OD)	5	21	52.34	<0.001
Nasal discharge (ND)	5	110	0.49	0.785
Conjunctivitis (CON)	5	21	49.01	<0.001
Depression (DEP)	5	21	4.12	0.009
Total clinical scores (TCSs)	5	21	40.00	<0.001

**Table 4 animals-15-00205-t004:** Tests from LMMs of effect of day on activity level (from FitBark) and distance travelled (from FitBark and TrackLab). Columns show numerator degrees of freedom (ndf), denominator degrees of freedom (ddf), F statistics, *p* values and variance components for batches, hens and batch × day compared to residual variance, with variance components divided by residual variance shown in brackets.

	Effect of Day				Variance Components (Variance Component/Residual Variance)			
	ndf	ddf	F Statistic	*p* Value	Batch	Hen	Batch × Day	Residual
Activity level (FitBark)	10	189	7.87	<0.001	0.1227 (0.12)	0.5023 (0.49)	0.0000 (0.00)	1.03
Distance travelled (FitBark)	10	189	7.28	<0.001	48,200 (0.06)	525,600 (0.60)	0.0006 (0.00)	868,800
Distance travelled (TrackLab)	10	39	4.76	<0.001	653,003 (3.97)	267,253 (1.62)	124,503 (0.76)	164,579

**Table 5 animals-15-00205-t005:** Tests from LMMs of effect of day on proportions of behaviours (angular transformed) per day from video scans. Columns show numerator degrees of freedom (ndf), denominator degrees of freedom (ddf), F statistics and *p* values.

	Effect of Day			
	ndf	ddf	F Statistic	*p* Value
Standing	10	42	1.91	0.070
Sitting	10	40	2.13	0.044
Walking	10	40	0.95	0.493
Foraging	10	42	2.21	0.036
Preening	10	40	1.41	0.210
Running	10	43	0.69	0.724
Drinking	10	40	0.56	0.833
Feeding	10	40	3.28	0.004
Pecking	10	42	0.85	0.584
Dustbathing	10	42	0.65	0.755
Nesting	10	40	0.75	0.674
Flapping	10	209	0.90	0.534
Active behaviours *	10	40	2.54	0.018

* Active behaviours: all behaviours except for standing, sitting, nesting.

**Table 6 animals-15-00205-t006:** Tests from LMMs of effect of day on proportions of total visit duration for locations (angular transformed) per day measured by TrackLab. Outer zone is not shown, as this is inversely related to the inner zone. Columns shown are numerator degrees of freedom (ndf), denominator degrees of freedom (ddf), F statistics and *p* values.

	Effect of Day			
	ndf	ddf	F Statistic	*p* Value
Inner zone	10	42	0.55	0.840
Litter zone	10	41	6.45	<0.001
Drinker	10	40	1.71	0.109
Feeder	10	40	0.82	0.609
Nest boxes	10	40	0.72	0.695
Perch	10	40	2.19	0.037

**Table 7 animals-15-00205-t007:** Tests from LMMs of effect of day on proportions of counts of hen locations (angular transformed) per day measured by video scanning. Outer zone is not shown, as this is inversely related to the inner zone. Perch is not shown as there were no observed visits to the perch. Columns shown are numerator degrees of freedom (ndf), denominator degrees of freedom (ddf), F statistics and *p* values.

	Effect of Day			
	ndf	ddf	F Statistic	*p* Value
Inner zone	10	40	1.97	0.062
Litter zone	10	40	2.71	0.012
Drinker	10	40	0.78	0.645
Feeder	10	40	1.47	0.185
Nest boxes	10	40	0.76	0.662

**Table 8 animals-15-00205-t008:** Tests and coefficients (gradients) from LMMs for effect of total clinical scores (TCSs) on activity level (from FitBark) and distance travelled (from TrackLab) for days 2–12. Columns show numerator degrees of freedom (ndf), denominator degrees of freedom (ddf), F statistics, *p* values and the ratio of variance components for batches, hens, days and batch × day compared to residual variance, with variance components divided by residual variance shown in brackets. Variance components from LMMs are also shown for the same models without TCS in fixed effects.

	Effect of TCS					Variance Components (Variance Component/Residual Variance)				
	ndf	ddf	F Statistic	*p* Value	Gradient (SE)	Batch	Hen	Day	Batch × Day	Residual
Activity level (FitBark)	1	19	77.43	<0.001	−0.315 (0.036)	0.0592 (0.07)	0.7220 (0.81)	0.0231 (0.03)	0.0000 (0.00)	0.8934
Activity level (FitBark)	(no fixed effects)					0.1230 (0.12)	0.5015 (0.48)	0.3564 (0.34)	0.0000 (0.00)	1.0349
Distance travelled (TrackLab)	1	146	47.64	<0.001	−144 (21)	639,686 (4.20)	328,498 (2.16)	0 (0.00)	128,284 (0.84)	152,330
Distance travelled (TrackLab)	(no fixed effects)					653,036 (3.97)	267,189 (1.62)	124,885 (0.76)	124,570 (0.76)	164,563

## Data Availability

The results of this study were derived from data which are stored at: https://zenodo.org/records/14161147.

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
