# Peer review of "Do Activity Sensors Identify Physiological, Clinical and Behavioural Changes in Laying Hens Exposed to a Vaccine Challenge?"

_animals, 2025, doi:10.3390/ani15020205_

Round 1
Reviewer 1 Report
Comments and Suggestions for Authors
The reported research addresses a significant animal welfare challenge by exploring ways to improve health monitoring. It investigated the use of wearable sensor technology to monitor laying hens, provides valuable insights into the potential of using activity data for identifying sickness symptoms in real-time. The study used multiple data sources such as physiological measurements, clinical symptom scoring, and video observations, to ensure a thorough evaluation of hens’ responses. Correlations between physiological, behavioral, and sensor data enhanced the study's findings.
The research is novel, however, the following concerns need to be addressed to ensure the broader impacts of the research findings:
1. Small sample size (20 hens) limit the generalizability of the results to larger commercial flocks. Including a larger sample size across multiple flocks would enhance the study's validity and applicability.
- The transition from a smaller home room to the larger experimental room appears to have influenced early data trends. A longer acclimatization period could improve the reliability of baseline measurements.
- As discussed in the manuscript, calibration issues led to concerns about the uniformity of sensor performance. It is recommended to include a validation phase for all sensors prior to deployment to ensure uniformity in data collection. In addition, it is needed to address the limitations of TrackLab by calibrating it to better differentiate between zones and activities
- While behavioral and physiological changes were observed, the manuscript notes difficulty in attributing these changes solely to the vaccine challenge as opposed to temporal effects or environmental adjustments. As a matter of fact, information about environmental conditions are missing. Since the experiments were conducted in June-August, were there any heat stress effects on hens? It is well known that heat stress could also affect animal activity levels.
- Water consumption could be a significant indicator of health, therefore, lack of water consumption data prevents a complete analysis of drinking behaviors.
Author Response
May we extend our grateful thanks to all reviewers for taking the time to read and comment on our draft paper.
We have changed the title to better reflect the data/analysis. We have changed ‘symptoms’ to ‘signs’ as in veterinary medicine, ‘symptoms’ is not used.
Comment 1.
Small sample size (20 hens) limit the generalizability of the results to larger commercial flocks. Including a larger sample size across multiple flocks would enhance the study's validity and applicability.
Response 1.
While we agree that what happens in an overall small group of hens is not directly translatable to commercial flocks, next steps would be to increase group sizes. However, despite this, significant effects were found in key measures. We have added to the conclusion at lines 722-724 that more/ larger studies are needed.
Comment 2.
The transition from a smaller home room to the larger experimental room appears to have influenced early data trends. A longer acclimatization period could improve the reliability of baseline measurements.
Response 2.
Thanks for your comment, we agree to a degree, although correlations between activity and total clinical scores (TCS) were strong. We have mentioned this in the Conclusion at lines 711-714.
Comment 3.
As discussed in the manuscript, calibration issues led to concerns about the uniformity of sensor performance. It is recommended to include a validation phase for all sensors prior to deployment to ensure uniformity in data collection. In addition, it is needed to address the limitations of TrackLab by calibrating it to better differentiate between zones and activities
Response 3.
Noted, thank you for your comment. Further detail on TrackLab not distinguishing bird height has been added to lines 204-205, however it is accurate to a location of ±15 cm as already stated in section 2.3.3. We explained why we did not validate either sensor in lines 573-574 but what problems were experienced. We also pointed out the lack of discrepancy by TrackLab when we discussed the perch values in lines 697-704.
Comment 4.
While behavioral and physiological changes were observed, the manuscript notes difficulty in attributing these changes solely to the vaccine challenge as opposed to temporal effects or environmental adjustments. As a matter of fact, information about environmental conditions are missing. Since the experiments were conducted in June-August, were there any heat stress effects on hens? It is well known that heat stress could also affect animal activity levels.
Response 4.
Environmental factors (such as relative humidity and temperature of experimental room) were controlled and kept constant by the automated management system. We recorded them once a day. We have now added mean ±SD (max) and mean ±SD (min) temperatures, and same for humidity, over the study period, in lines 137-139.
As well, we have now added formal tests from LMMs of the association between the sensor measurements and total clinical scores, so we are more reassured this was a response to vaccine challenge (lines 477-495 and Table 8).
Comment 5.
Water consumption could be a significant indicator of health, therefore, lack of water consumption data prevents a complete analysis of drinking behaviors.
Response 5.
We did not record water intake in this study, so we cannot include any water consumption information. We have noted your comment for any further studies we might undertake, thank you.
Reviewer 2 Report
Comments and Suggestions for Authors
The study conducted a series of experiments involving vaccine injections for laying hens. The workload for these experiments was considerable and demonstrated some innovation. If it were possible to replace the vaccine injections with virus injections, better results might be achieved. However, there are significant issues with this manuscript:
â‘ The theme of the paper is unclear. Wearable sensors have been used in animal research for many years. The title of the paper suggests that it aims to explore whether sensors can detect behavioral changes caused by vaccines. However, the abstract mentions uncertainty about whether the behavioral changes are due to the vaccine injection. As a result, the research content and the title of the paper are inconsistent, and the conclusion drawn from the study remains unclear.
â‘¡Regardless of whether a vaccine is injected, sensors can be used to monitor the activity of laying hens. What is the significance of mentioning vaccine injection? Could it aid in diagnosis?
â‘¢Line 84 states that the purpose of this study is to investigate whether sensors can detect if laying hens are ill. However, the paper does not include a process for pathological examination of the hens. Therefore, it is impossible to determine whether the behavioral abnormalities observed in the hens are due to a mild illness caused by the vaccine injection, which impacts their behavior.
â‘£Line 123 mentions, "A pilot study was conducted prior to the current study to determine which vaccine could induce measurable respiratory symptoms and to identify the period of immune response." Has this pilot study been published? Where is it cited?
⑤Each test group consists of only 4 hens, which reduces the persuasive power of the results. For small animals like chickens, it is recommended to have at least 10 animals per group to ensure more reliable and statistically significant findings.
â‘¥The Conclusion section mentions that the results obtained from the sensors used in this study cannot even guarantee earlier detection than humans. Therefore, what is the significance of the high cost of the sensors? This should be discussed in more detail.
Comments on the Quality of English LanguageThe English could be improved to more clearly express the research.
Author Response
May we extend our grateful thanks to all reviewers for taking the time to read and comment on our draft paper.
We have changed the title to better reflect the data/analysis. We have changed ‘symptoms’ to ‘signs’ as in veterinary medicine, ‘symptoms’ is not used.
Comments 1:
The theme of the paper is unclear. Wearable sensors have been used in animal research for many years. The title of the paper suggests that it aims to explore whether sensors can detect behavioral changes caused by vaccines. However, the abstract mentions uncertainty about whether the behavioral changes are due to the vaccine injection. As a result, the research content and the title of the paper are inconsistent, and the conclusion drawn from the study remains unclear.
Responses 1:
We conducted additional LMM analysis that demonstrates that changes in activity/distance travelled are highly associated to total clinical scores (3.6. the effect of total clinical score on sensor measurements – activity and distance, lines 477-495 and Table 8). This has reduced our reservations about potential day effects. The results from this analysis were added in abstract (lines 33-35) and discussion at lines 602-603, 618-619 and conclusion at lines 710-717.
Comments 2:
Regardless of whether a vaccine is injected, sensors can be used to monitor the activity of laying hens. What is the significance of mentioning vaccine injection? Could it aid in diagnosis?
Responses 2:
There are few papers that use sensors to identify changes in activity and behaviour of animals caused by disease. While this is the long-term goal (i.e., could sensors detect changes due to disease such as Avian Influenza), we used a live vaccine as an ethical method to induce disease-like symptoms, as opposed to infecting birds with an actual disease. We have clarified in lines 83-87 that the use of vaccines is as a proxy for disease infection.
Comments 3:
Line 84 states that the purpose of this study is to investigate whether sensors can detect if laying hens are ill. However, the paper does not include a process for pathological examination of the hens. Therefore, it is impossible to determine whether the behavioral abnormalities observed in the hens are due to a mild illness caused by the vaccine injection, which impacts their behavior.
Responses 3:
Please see point 2 above; in addition, we mention that studies using sensors on animals that are likely to develop disease could validate their use further. (lines 722-724)
Comments 4:
Line 123 mentions, "A pilot study was conducted prior to the current study to determine which vaccine could induce measurable respiratory symptoms and to identify the period of immune response." Has this pilot study been published? Where is it cited?
Responses 4:
The pilot study has not been published yet. We have clarified this in line 128.
Comments 5:
Each test group consists of only 4 hens, which reduces the persuasive power of the results. For small animals like chickens, it is recommended to have at least 10 animals per group to ensure more reliable and statistically significant findings.
Responses 5:
We agree that, ideally, for hens to behave normally, larger groups per batch could have been advantageous for reliable results. However, more birds would require more expensive activity monitoring sensor tags and would more than double the data produced (and taken considerably more time to extract behaviour data from video). We must balance ideals with practicality.
Comments 6:
The Conclusion section mentions that the results obtained from the sensors used in this study cannot even guarantee earlier detection than humans. Therefore, what is the significance of the high cost of the sensors? This should be discussed in more detail.
Responses 6:
As mentioned in the introduction chapter, sensors can be used to monitor health conditions and behaviour changes of individual animals in large groups. Although detection earlier than human observation has not been clearly demonstrated in this research, sensor data were highly associated with clinical scoring by human observation. Sensors that identify individual changes could be useful in terms of animal welfare and health management as an alternative to human observation.
Reviewer 3 Report
Comments and Suggestions for Authors
Review Report
This manuscript addresses the application potential of activity sensors be used in early indicators of behavior change in laying hens exposed to a vaccine challenge. The study demonstrates the wearable sensors detected changes in activity levels, distance travelled and location of individual hens. Sensor technology to detect changes in behaviors and movements of animals may be useful for monitoring animal health and welfare, but certain aspects require improvement. Below, I provide detailed comments and suggestions for improvement.
The keyword used in the article is “housing design”, but this is not the main focus of the research in this article and needs to be adjusted.
In the experimental design, five groups of replicated experiments were carried out successively, so the age of chickens in each group at the time of the experiment was different, how to avoid the following two problems: firstly, whether to consider the effect of different growth ages of chickens in different replicated groups on the results of the experiment; secondly, the experiment did not set up a blank control group, so how to avoid the exclusion of the chickens due to the changes in chickens' physiological and behavioral characteristics brought about by environmental factors.
A mixed effects model was used in the statistical analysis, and the model formula should be provided.
There is a need to harmonize the fonts of the horizontal and vertical axes of the graphs in the article, e.g., (a) and (b) in Figure 2.
Author Response
May we extend our grateful thanks to all reviewers for taking the time to read and comment on our draft paper.
We have changed the title to better reflect the data/analysis. We have changed ‘symptoms’ to ‘signs’ as in veterinary medicine, ‘symptoms’ is not used.
Comments 1:
The keyword used in the article is “housing design”, but this is not the main focus of the research in this article and needs to be adjusted.
Responses 1:
We have removed ‘housing design’ and added ‘clinical signs’ and ‘vaccine challenge’.
Comments 2:
In the experimental design, five groups of replicated experiments were carried out successively, so the age of chickens in each group at the time of the experiment was different, how to avoid the following two problems: firstly, whether to consider the effect of different growth ages of chickens in different replicated groups on the results of the experiment; secondly, the experiment did not set up a blank control group, so how to avoid the exclusion of the chickens due to the changes in chickens' physiological and behavioral characteristics brought about by environmental factors.
Responses 2:
Environmental factors (such as relative humidity and temperature of experimental room) were controlled and kept constant by the automated management system. We checked and recorded them in the logbook every day (once/day). Batch and day within batch were included in the random effects of the LMMs which would mop up any age effect; but there was no evidence of trend with increasing batch numbers anyway.
Comments 3:
A mixed effects model was used in the statistical analysis, and the model formula should be provided.
Responses 3:
These have now been added to the section 2.5 (Statistical analysis) – 3 formulae as there are 3 LMMs.
Comments 4:
There is a need to harmonize the fonts of the horizontal and vertical axes of the graphs in the article, e.g., (a) and (b) in Figure 2.
Responses 4:
This has now been corrected; however, we will work with the editors/typesetters of the journal if any other font discrepancies are discovered, thank you.
Round 2
Reviewer 1 Report
Comments and Suggestions for Authors
The manuscript is publishable, but it may be improved with the following minor revisions:
1. addressing potential limitations of the experimental design, such as reliance on a single observer for behavior and clinical assessments
2. elaborating how the findings could translate to practical applications in commercial poultry systems
3. Providing stronger recommendations for future research, especially focusing on refining sensor technology or integrating it with other monitoring systems
Author Response
May we extend our grateful thanks to all reviewers for taking the time to read and comment on our draft paper.
Comment 1.
Addressing potential limitations of the experimental design, such as reliance on a single observer for behavior and clinical assessments
Response 1.
Thanks for your comment.
We did not change the manuscript but I would explain that
Behavioural observation and clinical assessment by a single observer are quite typical in the field of animal research and help to avoid inter-observer reliability issues. As a veterinarian, I have assessed various clinical signs and behavioural changes by infectious animal diseases. We created the clinical scoring systems based on main clinical signs caused by Infectious Laryngotracheitis virus and by referring to the clinical signs observed in pilot study.
Comment 2.
Elaborating how the findings could translate to practical applications in commercial poultry systems
Response 2.
Thanks for your comment.
'Sensors may have further use in commercial poultry systems, by monitoring resource use, alerting producers to subtle changes in their behaviour (for example, predicting smothering events), and predicting the effects of stressful events such as aggressive feather pecking and keel bone damage.'
We have mentioned this in the Conclusion at lines 721-724.
Comment 3.
Providing stronger recommendations for future research, especially focusing on refining sensor technology or integrating it with other monitoring systems
Response 3.
Thanks for your comment.
'For sensors to be practical in a commercial environment, they must be smaller, less expensive, easily applied and robust, with a long-life or self-charging power source that does not need recharging within a flock's lifetime. Sensors that are similar (in size and application) to wing tags or leg rings would be pragmatic. In addition, data must be easy to access and interpret for farm staff. Sensor technology will continue to contribute to Smart Farming and Precision Agriculture, ideally with animal welfare at its core.'
We have mentioned this in the Conclusion at lines 726-732.